# Biological Activities of Aqueous Extracts from Carob Plant (*Ceratonia siliqua* L.) by Antioxidant, Analgesic and Proapoptotic Properties Evaluation

**DOI:** 10.3390/molecules25143120

**Published:** 2020-07-08

**Authors:** Siwar Ben Ayache, Emna Behija Saafi, Fathi Emhemmed, Guido Flamini, Lotfi Achour, Christian D. Muller

**Affiliations:** 1Research Laboratory “Bioresources, Integrative Biology & Valorization”, Higher Institute of Biotechnology of Monastir, University of Monastir, Monastir 5000, Tunisia; saafi_behija@yahoo.fr (E.B.S.); lotfiachour@yahoo.fr (L.A.); 2Institut Pluridisciplinaire Hubert Curien, UMR 7178 CNRS, Faculté de Pharmacie, Université de Strasbourg, 67401 Illkirch, France; fathi.emhemmed@iphc.cnrs.fr; 3Dipartimento di Farmacia, Università di Pisa, Via Bonanno 6, 56126 Pisa, Italy; guido.flamini@unipi.it; 4Centro Interdipartimentale di Ricerca “Nutraceutica e Alimentazione per la Salute” Nutrafood, Università di Pisa, 56124 Pisa, Italy

**Keywords:** *Ceratonia siliqua* L., decoction, pulps, seeds, rob, volatiles, phytochemicals, antioxidant, analgesic, proapoptotic activities

## Abstract

The present work describes the volatile compounds profile and phytochemical content of *Ceratonia siliqua* L. Fifty different components have been identified. Among them, three constituents are shared i.e., 2-methlybutanoic acid, methyl hexanoate and limonene by different common carob preparations: pulp decoction (PD), seeds decoction (SD) and Rob, a sweet syrup extracted from the pulp of the carob pod. Each extract exhibits different volatile aromatic emission profiles. The antioxidant activity of the extracts was evaluated using three methods, DPPH, ABTS and FRAP, producing a dose-dependent response. The IC_50,_ when determined by FRAP, gave the lowest values (0.66 ± 0.01, 0.73 ± 0.05 and 0.55 ± 0.00 mg/mL PD, SD and Rob, respectively). The nociception essay, after intraperitoneal injection of acetic acid in mice, demonstrated that Rob, pulp and seeds decoction extracts showed an efficient inhibition of writhes over time, with persistence over 30 min. The SD decoction revealed the highest efficacy in decreasing the writhing reflex (90.3 ± 1.2%; *p* < 0.001). Furthermore, the proapoptotic activity of SD against three human cell line, THP-1, MCF-7 and LOVO, evaluated by flow cytometry, showed a significantly stronger proapoptotic activity on colon cancer (LOVO) than on the other cell lines, a phenomenon known as phenotypic selectivity.

## 1. Introduction

Consumers are increasingly aware of the benefits of foods and plants that provide not only essential nutrients but also phytochemicals essential for health. Among all the bioactive compounds identified in plants, some of them, such as phenols, flavonoids, coumarins and curcuminoids can be absorbed by most of the cells through the blood circulation. This can explain the beneficial effects of many plants on several pathologies as well as disease prevention thanks to their antioxidant, anti-inflammatory and antitumor properties [1].

Depending on their concentrations, reactive oxygen species (ROS) can play either a key physiological role or a toxic effect. In a normal physiological state, ROS are generated in small amounts to regulate apoptosis or activate transcription factors acting as secondary messengers in the intracellular signaling pathway. Excessive generation of ROS becomes pathologic by activating the expression of gene coding for proinflammatory cytokines. Their unstable nature made them very reactive towards biological substrates and induced deleterious oxidative modifications potentially involved in several pathologies, particularly in cancer generated by DNA mutations [2]. The cumulative production of ROS within the cell, whatever endogenous or exogenous, induces cellular senescence and apoptosis. Apoptosis is associated with an altered redox regulation of intracellular signaling cascades involved in carcinogenesis [3]. A multifaceted bacteria community colonizes the gastrointestinal tract and is recognized today as a contributor to digestive tract health and mainly colorectal cancer prevention [4]. Cell infiltration and intestinal mucosal abscesses characterize chronic inflammation, namely ulcerative colitis. The infiltration of neutrophils, lymphocytes and eosinophils in an inflammatory colon is explained by the production of large amounts of ROS. These infiltrated plasma cells lead to oxidative stress and proteolytic enzymes, triggering small intestinal mucosal necrosis and ulceration [5].

Tunisian flora is well known for its plant biodiversity [6], among which *Ceratonia siliqua* L. (commonly named Carob), an evergreen tree belonging to the Fabaceae family and largely cultivated in the Mediterranean basin [7]. Carob is used in several industries: food, pharmaceutics and cosmetics [8]. Diverse studies have shown that carob has several biological activities, i.e., antioxidant, antitumor and antibacterial ones [9,10]. Carob pods and seeds are often used in traditional Tunisian medicine as analgesic, anticonstipation, antiabsorptive of glucose, in gastrointestinal propulsion and antidiarrhea activities [11]. Phenolic compounds of an aqueous carob extract modulate gene expression related to the H_2_O_2_ genotoxic impact on colon adenoma cells [12]. Since the aqueous extract has shown a very high antioxidant and radical scavenging activities in vitro [13], this work investigated only water carob extracts. Molasses or syrup is a traditional Tunisian food commonly known as “Rob”. Rob helps to conserve seasonal food and is widely consumed in Mediterranean countries e.g., Lebanon, Tunisia and Turkey [14,15,16].

According to the Food and Agriculture Organization of the United Nations (FAO, 2018), the production of *C. siliqua* beans in Tunisia decreased from 1000 tons in 2000 to 847 in 2017. In the Tunisian flora, the Carob tree is considered a vulnerable species whose transition to endangered species is considered today [6]. In this context, this study aims to analyze the volatile and nonvolatile phytochemicals of Tunisian varieties of Carob and to study antioxidant, analgesic and proapoptotic activities on cancer cells of leukemia (THP-1), breast (MCF-7) and colon (LOVO), to contribute to the valorization of the Tunisian flora, in order to encourage the production and use of locust beans in the biotechnological fields as a natural bioactive resource for food and pharmaceutical industries.

## 2. Materials and Methods

### 2.1. Plant Material

*Ceratonia siliqua* L. pods were harvested in Teboulba (Monastir Governorate, 35°38′31″ N, 10°57′57″ E), Tunisia, during August 2017 at maturity stage. The plant material was equally divided into two samples: pods with seeds for Rob preparation and pods without seeds for the decoctions. Pulps and seeds were dried and ground to a fine powder with an electric mixer. The powdered samples were then stored at −20 °C until extraction by decoction.

### 2.2. Preparation of the Extract

According to the traditional method, Rob was prepared in three main steps, namely: extraction, filtration and concentration. The process consists of washing and sun-drying the carob pods (pulps + seeds) until complete dehydration, followed by crushing into a traditional mortar. Six kilograms of ground pods were mixed with 15 L of water and kept overnight. The next day, the maceration was filtered through a cloth. The filtrate was then boiled for about 6 h until a syrup consistency was obtained, then conserved in glass bottles at 4 °C.

### 2.3. Methods

#### 2.3.1. Volatile Compound Analysis

The aroma volatiles spontaneously emitted by the samples were determined using solid-phase microextraction (SPME) coupled with Gas Chromatography-Mass Spectrometry (GC–MS).

In brief, 5 g of each sample was individually put into 4 mL glass vials (up to 1/3 of their total volume) and left to equilibrate at room temperature for 30 min. Supelco SPME devices coated with polydimethylsiloxane (PDMS, 100 μm) were used to sample the headspace of the samples. SPME sampling was performed at room temperature using the same fiber, preconditioned according to the manufacturer instructions, for all the analyses. After 30 min of equilibration time, the fiber was exposed to the headspace for 30 min. Once sampling was finished, the fiber was withdrawn into the needle and transferred to the injection port of the GC–MS system. Quantitative comparisons of relative peaks areas were performed between the same chemicals in the different samples.

GC–MS analyses were performed with an Agilent 7890B gas chromatograph (Agilent Technologies Inc., Santa Clara, CA, USA) equipped with an Agilent DB-5MS (Agilent Technologies Inc., Santa Clara, CA, USA) capillary column (30 m × 0.25 mm; coating thickness 0.25 μm) and an Agilent 5977B quadrupole mass detector (Agilent Technologies Inc., Santa Clara, CA, USA). The analytical conditions were as follows: injector and transfer line temperatures 220 and 240 °C, respectively; oven temperature programmed from 60 to 240 °C at 3 °C/min; carrier gas helium at 1 mL/min; injection of 1 μL (0.5% HPLC grade *n*-hexane solution); splitless injection. Identification of the constituents was based on a comparison of their linear retention indices relative to the series of *n*-hydrocarbons. Computer matching was also used against commercial (NIST 14 and ADAMS 2007) and laboratory-developed mass spectra libraries built up from pure substances and components of commercial essential oils of known composition and MS literature data [17,18,19,20,21,22].

#### 2.3.2. Phytochemical Content

The anthocyanin content was determined according to the method of Giusti and Wrolstad [23] and expressed as mg cyanidin 3-glucoside equivalents (Cy E) per 100 g of dry weight. The total phenol content of carob pulps seeds and Rob was estimated by the colorimetric method of Folin–Ciocalteu according to Al-Farsi and al. [24]. The total flavonoids content was determined by a colorimetric method according to Reis and al. [25] and expressed as g of catechin equivalents per 100 g of dry weight. The condensed tannins were estimated according to Julkunen-Titto et al. [26] and expressed as g of catechin equivalents per 100 g of dry weight.

#### 2.3.3. Antioxidant Activity

The antioxidant effectiveness of *C. siliqua* decoctions was assessed using three different tests: ferric reducing antioxidant power (FRAP), DPPH radical scavenging assay and ABTS scavenging activity. All measurements were carried out according to Pinela and al. [27] and to Barros and al. [28], respectively.

#### 2.3.4. Acute Toxicity

Swiss albino mice of both sexes weighing 18–25 g, were obtained from Pasteur Institute (Tunis, Tunisia). The animals were housed under standard laboratory conditions with a 12 h light–dark cycle at constant temperature (22 ± 2 °C) and humidity (55 ± 5%) and kept on a commercial diet and tap water provided *ad libitum*. The animals were handled according to the guidelines of the Tunisian Society for the Care and Use of Laboratory Animals, and the study was approved by the University of Monastir Ethical Committee (Approval No: CER-SVS 007/2020 ISBM).

The acute toxicity was evaluated using groups of six Swiss Albino mice. One group serves as a control and receives 0.9% NaCl (10 mL/kg) intraperitoneally (i.p.), the remaining group was treated with increasing doses of pulp, seeds and Rob at four doses: 0.3, 1.0, 1.5 and 2.0 g/kg (i.p.). The mortality rate within a 48 h period was determined as LD_50_ estimated according to the method of Miller and Tainter [29]. Accordingly, nonacute toxicity doses are chosen for pharmacological evaluations.

#### 2.3.5. Analgesic Activity

Antinociceptive activity was based on the procedure described by Koster and al. [30]. The nociception was induced by i.p. injection of acetic acid. The animals were divided into 14 groups of six each. The control group was treated subcutaneously with 10 mL/kg of 0.9% NaCl, a second group with 200 mg/kg of lysine acetyl salicylate. The latest groups were treated subcutaneously with an aqueous decoction of pulps, seeds extract and Rob at different doses (50, 100, 150, 200 mg/kg). Thirty minutes after administration, all groups received i.p. 10 mL/kg of 1% acetic acid. The number of abdominal writhes, indicators of pain, was counted for 30 min. Thus, the antinociceptive activity was expressed as a percentage of inhibition of abdominal constriction.

#### 2.3.6. Cell Cultures

All human cell lines were purchased from ATCC (LGC Standards, Molsheim, France). Human mammary (MCF-7, ATCC^®^ HTB-22) and colon LoVo adenocarcinoma (ATCC^®^ CCL-229^™^) cell lines were maintained in DMEM high glucose medium (Dominique Dutscher, 67,172 Brumath, France, Cat No. L0102-500), while human acute monocytic leukemia cell line (THP-1, ATCC^®^ TIB-202) was maintained in RPMI-1640 Medium (ATCC^®^ 30-2001™, LGC Standards, Molsheim, France), supplemented with 10% (*v*/*v*) heat inactivated fetal bovine serum (FBS, Life Technologies, Paisley, UK, Cat No. 10270-106) and 1% (*v*/*v*) penicillin–streptomycin (10,000 units/mL and 10,000 µg/mL, Life Technologies, Paisley, UK, Cat No. 15140-122). Cells were kept at 37 °C in a humidified atmosphere containing 5% (*v*/*v*) CO_2_ during their exponential growing phase and in the course of incubation with investigated compounds. Before confluence, adherent cells were trypsinized and subcultured twice a week.

#### 2.3.7. Proapoptotic Activity

Cells were incubated for 24 h at 37 °C, in presence of the different carob aqueous extracts at increasing concentrations, i.e., 1000, 500, 250, 150 and 75 µg/mL per well containing 10^5^ cells/mL. LoVo and MCF-7 cells were left overnight to settle, while treatment of THP-1 cells started 2 h after seeding. For nonadherent THP-1 cells PBS, Annexin V-FITC and propidium iodide (PI) were added to wells right after incubation times. For adherent LoVo or MCF-7 cells, a trypsinization protocol was applied each time prior to flow cytometry analysis. Previously removed supernatants with nonadherent apoptotic cells were returned to detached cells and stained with Annexin IV-FITC (ImmunoTools GmbH, Friesoy the, Germany, Cat No. 31490013) and propidium iodide (PI, Miltenyi Biotec Inc., Auburn, USA, Cat No. 130-093-233) in a volume of 2 µL each. A minimum of 5000 cells was acquired per sample and analyzed with the InCyte software. Apoptosis rates were assessed by capillary cytometry using Annexin V-FITC and PI according to the manufacturer recommendations. Gates were drawn around the appropriate cell population using forward scatter (FSC) versus side scatter (SSC) acquisition dot plot to exclude any debris. To discriminate between negative and positive events in the analysis, a nonstained control sample from each culture condition always accompanied acquisition of the stained cells to define their cut off. Negative control i.e., samples with cells without compounds but with the same amount of water as for diluted compounds, as well as positive control with 50 μM Celastrol, a natural pentacyclic triterpenoid (Enzo Life Sciences, Farmingdale, NY, US), were included in each experiment. Cytometers performances were checked weekly using the Guava easy Check Kit 4500–0025 (Guava/Luminex, Santa Clara, CA, USA). Cells were categorized according to Annexin-V-FITC (green fluorescence) and PI (red fluorescence) labeling on viable (double negative), preapoptotic cells (Annexin V-FITC single-stained cells), necrotic cells (PI single-stained cells), and cells in advanced phases of apoptosis (double-stained cells).

### 2.4. Statistical Analyses

All tests were run in 3 independent experiments (*n* = 3) and results expressed as mean values with standard deviation (±SD). The values were computed using the one-way analysis of variance (ANOVA) followed by Tukey’s test. Analyses were performed with SPSS v. 22.0 program (IBM Armonk, NY, USA) with *p* < 0.05 considered as statistically significant.

## 3. Results and Discussion

### 3.1. Volatile Compounds

Full characterization of the aromatic volatile compounds of *C. siliqua* performed by solid-phase microextraction coupled with GC-MS is shown in Table 1. Fifty compounds were identified, accounting for 98.9, 96.8 and 99.2% of total released compounds in carob pulps, seeds and Rob, respectively. Seeds were the sample that emitted the highest number of volatiles (34), followed by pulps (19) and Rob (11). Four different chemical classes were characterized, in particular monoterpene hydrocarbons, oxygenated monoterpenes, nonterpene derivatives and sulfur derivatives. Among nonterpenes, further subclasses can be individuated, i.e., nonterpene hydrocarbons, nonterpene carbonyl compounds (i.e., aldehydes, ketones, acids), nonterpene alcohols/phenols and nonterpene esters. Esters were the main volatile compounds for seeds and pulps, reaching 16.9% and 91.8%, respectively. Whereas, in Rob they reached very low percentages (0.2%). Furthermore, our analysis showed that, among the 15 nonterpene esters, only one was common to all the different samples, namely methyl hexanoate (7.8%, 78.6% and 0.2%, respectively in seeds, pulps and Rob). Further, they share other compounds such as 2-methylbutanoic acid (2.6, 4.7% and 6.3%, respectively) and the monoterpene hydrocarbon limonene (34.0%, 0.6% and 1.0%, respectively). The seeds exclusively emitted nonterpene hydrocarbons and terpenes, both hydrocarbons and oxygenated ones, with the exception of limonene. Volatile acids characterized mainly Rob, in particular isobutyric acid (79.5%), while pulps were mainly dominated by esters, like methyl hexanoate, reaching 78.6%. Hydrocarbons were detected exclusively in seeds. Our data also shows that pulps, seeds and Rob contain the highest amount of nonterpene derivatives, reaching the highest amount in pulps, seeds and Rob (97.8%, 46.2% and 94.1%, respectively. The analyses reveal that the samples of carob have clear differences in their emission profiles concerning proportions and kind of volatile components.

According to the literature, studies have identified different chemical classes of volatile compounds [31]. In this context, Krokou et al. [32] found 163 molecules as volatile components in Spanish carob deseeded pods, mainly aliphatic acids (77.5%) and aliphatic esters (10.52%). In contrast, a Tunisian report [33] has identified 25 different compounds in the essential oil of carob pods, with a complex mixture of hydrocarbons, terpenoids, esters, alcohols, ketones, fatty acids and aldehydes. The differences may be explained by the dissimilarity in cultivars and the ecological factors affecting the growth process, also because of the different types of extract and sampling techniques. According to Hanousek Čiča, the characterization of aroma compounds of carob pods macerates in hydroalcoholic base shows 27 components subdivided into 17 esters, four ketones, two acids and three alcohols, constituting of 70–85% of total aroma compounds [34]. Another study identified seven different volatile groups, with acids as the most abundant class (71%–77% in pods powder) [35]. Despite the extraction method, difference seems to be primarily due to ecological factors and to the extracted organ: pods with seeds or pods deseeded and to the type of investigated cultivars i.e., male, female or hermaphrodite [36]. In regards to “Rob”, there is a difference in the volatile profile of homemade Rob, characterized by fatty acids and terpenes, and the commercial Rob molasses characterized by terpenoids and siloxanes, the latter being non natural products [16]. The raw material as well as the manufacturing process can explain these differences.

### 3.2. Phytochemical Analysis of Carob

As shown in Table 2, the highest amount of anthocyanins was found in Rob with 188 ± 5 mg/100 g DW, followed by pulps and seeds extracts (158.1 ± 0.7 and 51.2 ± 0.1 mg/100 g DW, respectively). A similar trend was observed for the total phenols content, with 13 ± 0.8, 1.8 ± 0.1 and 0.9 ± 0.1 g EAG/100 g DW in Rob, pulps and seeds extract decoction, respectively. Rob also contains the highest amount of flavonoids (2.4 ± 0.1 g EC/100 g DW). The level of flavonoids in pulp decoction (PD) and seeds decoction (SD) extracts is around five times lower, with 0.6 ± 0.0 and 0.6 ± 0.1 g EC/100 g, respectively. For the condensed tannins, their amounts are 1.2 ± 0.1 g EC/100 g DW in Rob; 0.2 ± 0.0 g EC/100 g DW in SD extract and 0.3 ± 0.0 g EC/100 g DW in PD. Several studies have demonstrated that carob is rich in phenolic compounds, but anthocyanins are here reported for the first time. Our results are in good agreement with a previous report considering total phenols, flavonoids and condensed tannin in methanolic extracts of three different cultivars. The values ranged between 16.4 and 34.2 mg GE/g DW, 2.1 and 13.4 mg RE/g DW and 1.8 and 23.7 mg CE/g DW, respectively [9]. According to other works, carob pods show variations, according to the used solvent system, ranging from 7.1 to 382.0 mg EGA of phenolic content and 98.7 mg EC of flavonoids [37]. Decoctions of pulps and germ (part of seeds) contain 8.2 and 6.1 mg GAE/g DW of phenolic compounds [38]. Other investigations found smaller amounts of secondary metabolites in deseeded carob pods: 2.2 mg EAG/g and 0.27 mg EC/g DW of total polyphenols and total flavanols [39].

### 3.3. DPPH Scavenging Activity

To investigate the antioxidant potential of our extracts, DPPH scavenging activity was measured. Our data summarized in Figure 1, highlights the ability of phenolic carob extracts to reduce the free radical DPPH in a dose-dependent manner. The highest DPPH scavenging activity of decoction pulps extract was observed from 5 mg/mL, with 80.9 ± 0.6%, similar to that of the seeds extract (79.4 ± 0.9%). However, DPPH scavenging activity of Rob was higher than the other samples, starting at 2.5 mg/mL (80.9 ± 0.2%). This behavior can be explained by the highest amount of phenolic content of Rob (13.0 ± 0.8 g/100 g DW).

### 3.4. ABTS Scavenging Activity

According to the results shown in Figure 2, carob presents a dose-dependent ABTS^.^ scavenging activity. Rob was the most active one (2.5 mg/mL, with 79.1 ± 0.5%). This ability to reduce free ABTS radical was greatly increased at 5 and 10 mg/mL, with 96.4 ± 0.2 and 97.0 ± 0.1%, respectively. Very likely, this activity results from the high amount of total phenol contained in this sample. The potent ABTS radical scavenging of pulps and seeds decoction extracts reached 90.9 ± 0.2 and 95.8 ± 0.5% at 10 mg/mL, respectively.

### 3.5. Antioxidant Activity Evaluated by Three Methods (FRAP, DPPH and ABTS)

The antiradical activity of carob samples increases in a dose-dependent way. This antioxidant capacity can be evaluated by the determination of IC_50_ values corresponding to the amount of the sample needed to reduce 50% of free radicals. Higher values of IC_50_ indicate lesser effectiveness.

According to Table 3, the IC_50_ determined by FRAP of the different extracts shows lower IC_50_ values than DPPH and ABTS tests. Moreover, Rob has the highest antioxidant activity for FRAP and ABTS, denoted by the lowest IC_50_ (0.55 ± 0.0, 0.94 ± 0.1 mg/mL, respectively). Seeds decoction extract shows a IC_50_ value in FRAP (0.73 ± 0.1 mg/mL) slightly different from the other two methods, 0.86 ± 0.0 and 1.61 ± 0.1 mg/mL for DPPH and ABTS, respectively. Also, in the case of pulps decoction FRAP shows the highest antioxidant activity (IC_50_ = 0.66 ± 0.0 mg/mL), while DPPH and ABTS values were 1.04 ± 0.04 and 3.49 ± 0.00 mg/mL, respectively.

Thus, Rob reveals potent antioxidant activity as shown by FRAP and ABTS and a high antiradical capacity. This antioxidant activity may be attributed to the levels of total phenols, flavonoids, condensed tannin and anthocyanin (13.0 ± 0.8 g EAG, 2.4 ± 0.1 g EC, 1.2 ± 0.1 g EC and 188.5 ± 5.0 DW, respectively). Furthermore, phenolic compounds, depending on their different properties, contribute to the antioxidant activity in a dose-dependent manner until a maximum of activity [40]. Our work is in agreement with previous findings: the reducing power of carob extract was reported to range from 0.2 to 1.0 mg/mL and to be 1.5 mg/mL for DPPH scavenging activity [41]. In another study, the IC_50_ of the DPPH test of carob seeds acetone extract was around 160 µg/mL [42]. As described here, and also in a previous study, the power of reducing ferric ion to ferrous ion in FRAP was more pronounced than that observed for DPPH and ABTS radicals (0.9 mg/mL versus 57.5 and 78.8 mg/mL, respectively) [43]. Moreover, ABTS radical scavenging exhibited the highest IC_50_ in the different samples, as confirmed by a Turkish study where IC_50_ of carob seeds was about 14.01 mM/g DW for ABTS and 3.34 and 6.19 mM/g DW for DPPH and FRAP [44]. The decoction of seeds reveals an important antioxidant activity. Since seeds, pulps and Rob exhibit both volatile and nonvolatile compounds, the antioxidant activity may be attributable to both these types of substances, as demonstrated in a previous report. Some volatile compounds, such as hydrocarbons (unsaturated compounds) inhibited the oxidation processes by the transfer of hydrogen atoms [45]. Given that, NFκB, a regulator of cytokine secretion, becomes phosphorylated and thus activated through oxygen-derived free radicals [46].

### 3.6. Acute Toxicity

PD, SD and Rob at various doses did not show any behavioral changes, nor any morbid symptoms or death in mice, even at the highest dose. Thus, the LD_50_ value by intraperitoneal route cannot be determined as no lethality was observed up to 2000 mg/kg in mice.

### 3.7. Analgesic Activity

As summarized in Table 4, the number of writhing reflexes decreased significantly in the group that received lysine acetyl salicylate or different doses of the three carob preparations. The reduction of writhes by the administration of the seeds extract, at the dose of 50 mg/kg was more effective in reducing writhes than the reference drug dose (200 mg/kg). Furthermore, the effect of the seed extract was efficiently maintained over time. On the contrary, in the case of the reference drug, the pain inhibition did not persist over time, decreasing from 90.0 ± 0.9 to 16.1 ± 1.3% in the next 30 min. For Rob and decoction extract, an increase of the power of inhibition with persistence over time at the same dose of the reference drug (200 mg/kg) was observed. Concerning the seeds extract decoction, the doses of 200 mg/kg and 150 mg/kg showed a significant reduction of writhing (*p* < 0.05) over time from 48.1 ± 3.4 to 90.3 ± 1.1% and 55.7 ± 3.5 to 75.8 ± 1.6%, respectively. For Rob, the dose of 150 mg/kg displayed the most important antinociceptive activity (84 ± 2%) at 30 min. SD was the most efficient preparation for decreasing the writhing reflex for 30 min, followed by Rob, then pulp decoctions. These results may be explained by the fact that, as shown earlier, SD exhibits a potent antioxidant activity with 39 volatile compounds out of 50 components identified in all tested samples. It was recently described that such volatile compounds largely contribute to the analgesic effects by reducing the mRNA expression levels of COX-2, TNFα, IL-1β and IL-13 [47]. The analgesic process is related to the sensitization of nociceptors to prostaglandins. Thus, acetic acid causes irritations leading to stimulation of prostaglandins e.g., PGE2 and PG2α [48]. Moreover, terpenes and terpenoids inhibit several steps of inflammatory processes and decrease the levels of proinflammatory biomarkers, as NO, TNFα, interleukins and PGE 2 [49]. Hence, the analgesic effect of carob samples is probably due to inhibition of the action or the release of prostaglandins. Carob samples are rich in phenolic compounds, such as flavonoids, largely involved in nonspecific immunological responses, as well as acute inflammatory reactions [50]. To the best of our knowledge, this is the first time that an antinociceptive activity of carob is observed.

### 3.8. Carob Extracts Induce Apoptosis of Leukemic, Breast and Colon Cancer Cell Line Models

Being our carob extracts rich in antioxidant compounds, we supposed that these preparations could have anticancer activity. For this purpose, their ability to trigger apoptosis in different cancer cell lines was assessed. As evidenced in Figure 3, Figure 4, Figure 5, leukemic, breast and colon cancer cell lines show a dose-dependent proapoptotic response (% of apoptotic cells = Annexin V positive population + Annexin V/PI positive population). The proapoptotic activities on THP1 and MCF-7 cells for the three carob formulations never exceed 20% of the total population. Noteworthy, the seeds extract reveals to be always the most potent antiproliferative one. Regarding the proapoptotic effects of carob extracts on LoVo cells, the seeds decoction again displays the most important effect. Indeed, pulp and seeds extracts exhibited the highest phenotypic effect onto the colon LoVo adenocarcinoma, inducing up to 45% of apoptotic cells. To summarize, carob extracts are more effective on colon cancer (LoVo) with a phenotypic specificity when compared to the two other tested cell lines. This study represents the first demonstration of the anticarcinogenic potential of aqueous carob extracts (pulps and seeds) and of a byproduct (Rob). Hsouna et al., 2011 have reported that carob essential oil possesses antitumor activity with an IC_50_ of 800 µg/mL and inhibits cell death by 35.04% on MCF-7 at the dose of 250 µg/mL [33]. The difference of these results can be explained by the use of different extracts, containing completely different bioactive compounds, besides the use of a different assay for detecting living cells by MTT oxidation and not considering apoptotic cells. In accordance with our findings, carob pod extracts reveal a cytostatic activity by arresting breast cancer MDA-MB-231 cells in the S phase, with 30% of the cells in sub-G1 (apoptosis) at 40 mg/mL [51]. Nevertheless, the literature demonstrates that carob pods exhibit a strong antiproliferative capacity on several cancer cell lines, different from those investigated here. The carob extract pods have shown a high inhibition on human colon cell lines growth, but the effect on DNA-synthesis was less pronounced in HT29 adenocarcinoma than LT97 adenoma cells, certainly due to different growth kinetics [52]. The aqueous extract of carob induces a significant reduction of the number of human colon cancer HT29 cells at the concentration of 10 mg/mL [12]. At 20 µg/mL, *C. siliqua* leaf extract induces the phosphorylation of p38 MAPK and p53 proteins in HCT-116 and CT-26 colon cancer cells via the activation of caspase-9 in order to induce apoptosis. Carob leaf infusion extracts activate not only caspase-9 but also caspase-3 and the PARP pathway, so to reduce tumor growth on CT-26 in BALB/c, demonstrating that carob leaf in fusion has anticancer properties both in vitro and in vivo [53]. Since polyphenols are well described for their antioxidant capacity, Klenow and coll. investigated the ability of carob extract to protect LT97 and HT29 cells against H_2_O_2_ inducer of DNA damage [12]. Sugars, particularly sucrose and glucose, may mask the cytotoxicity of other bioactive components and contribute to the reduction of anticarcinogenic capacity of such extract [54]. This study found that carob reduces the growth of HT29 colon carcinoma and LT97 colon adenoma cells via the modulation of CAT and SOD2 gene expression, as well as the detoxification of ROS, which are associated with the colorectal carcinogenesis [12]. In order to determine an accurate IC_50_ value for seeds extracted on proapoptotic activity and intracellular ROS released by LOVO cells, we need to test lower concentrations of our extracts, to assess possible gene modulation expression related to the stress response.

## 4. Conclusions

Overall, our results reinforce the view that carob pulps, seeds and Rob may be implicated in several pathways, contributing to antioxidant activity by means of their bioactive components. According to our data, the seeds carob extract exhibits potent radical scavenging properties, resulting in analgesic activity. Most importantly, the reduction of the writhing reflex of the investigated extract is more effective than the control drug, which seems to be an attractive new approach for using carob extracts to avoid secondary effects of analgesic medicines. On the other hand, reactive oxygen species provoke many degenerative diseases as well as cancer. This statement prompted us to assess anticarcinogenic activity on THP1, MCF-7 and LoVo human cells. Our data demonstrate that carob decoction extract reveals an effective phenotypic proapoptotic capacity on colon cancer cell lines. Among the three types of carob extracts tested here, the seeds show the most potent anticarcinogenic properties on all the tested human cell carcinomas.

## Figures and Tables

**Figure 1 molecules-25-03120-f001:**
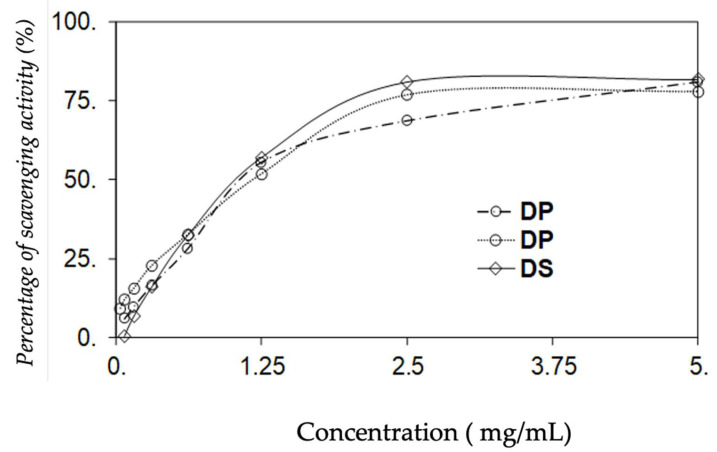
DPPH. scavenging activity based on the concentration of phenolic extracts of carob from Teboulba. DP: decoction pulps extract from Teboulba, DS: decoction seeds extract and Rob: traditional Tunisian preparation.

**Figure 2 molecules-25-03120-f002:**
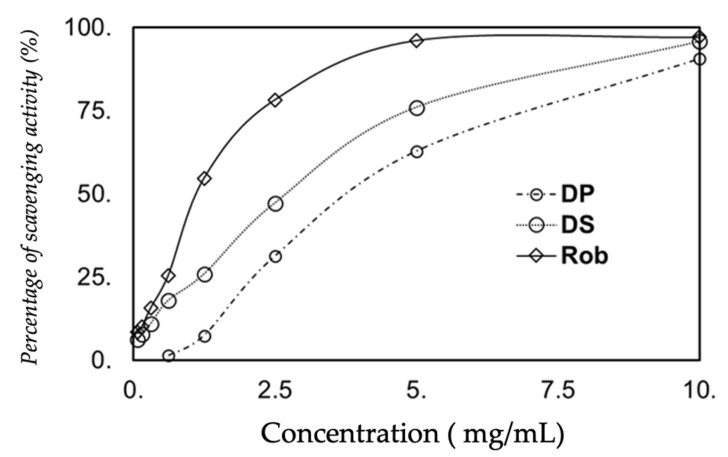
ABTS radical scavenging depending on concentrations of extracts (mg/mL) from Teboulba. DP: decoction pulps extract from Teboulba, DS: decoction seeds extract and Rob: traditional Tunisian preparation.

**Figure 3 molecules-25-03120-f003:**
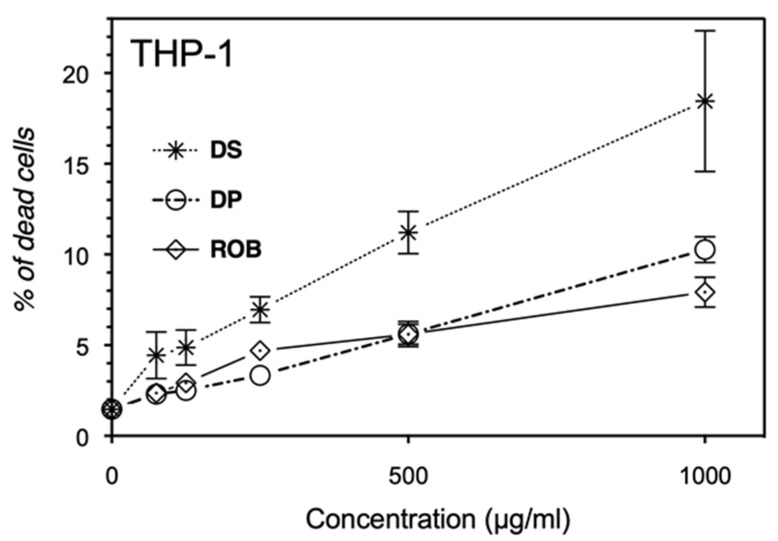
THP-1 monocytic leukemia cells treated with increasing doses of carob extracts. DS: Cells treated with seeds decoction extract; DP: Cells treated with pulp decoction extract and Rob: Cells treated with Rob. Results are expressed as mean ± SD with *n* = 3 independent experiments.

**Figure 4 molecules-25-03120-f004:**
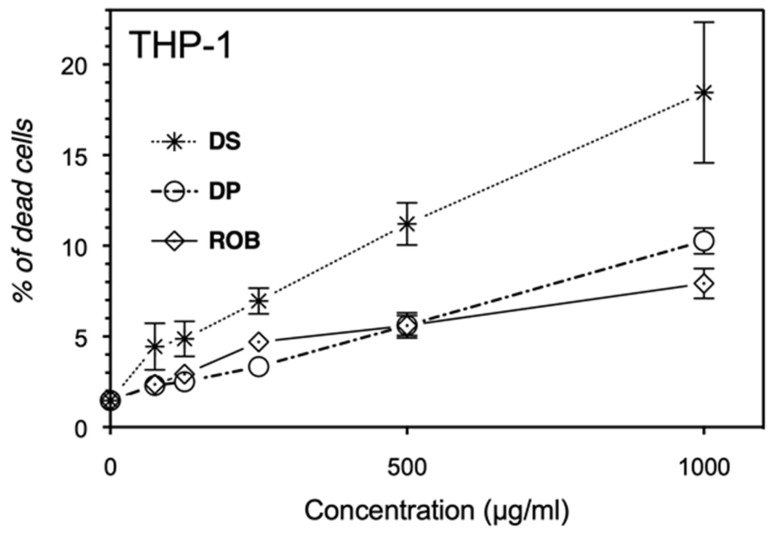
Breast cancer cells (MCF-7) treated with increasing doses of carob extracts. DS: Cells treated with seeds decoction extract; DP: Cells treated with pulp decoction extract and Rob: Cells treated with Rob. Results are expressed as mean ± SD with *n* = 3 independent experiments.

**Figure 5 molecules-25-03120-f005:**
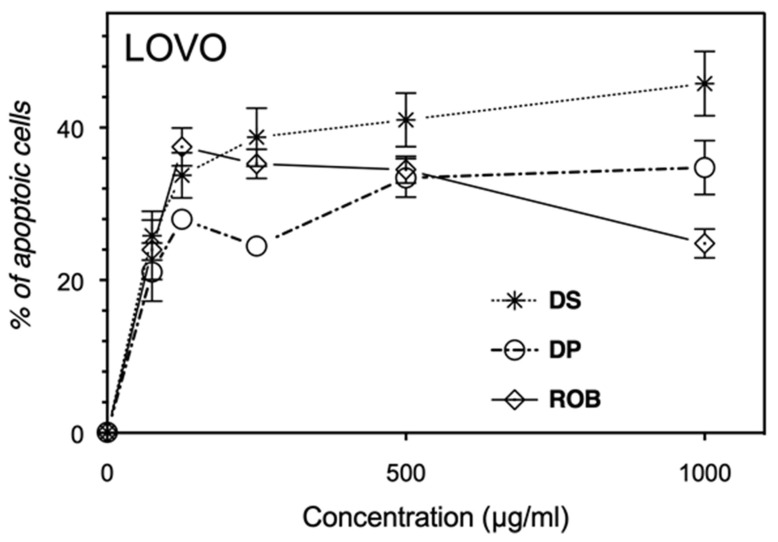
Adenoma colon cells (LOVO) treated with increasing doses of carob extracts. DS: Cells treated with seeds decoction extract; DP: Cells treated with pulp decoction extract and Rob: Cells treated with Rob. Results are expressed as mean ± SD with *n* = 3 independent experiments.

**Table 1 molecules-25-03120-t001:** Volatile compounds evaluated in *Ceratonia siliqua* L. pulps, seeds and Rob.

Constituents ^a^	l.r.i. ^b^	Seeds	Pulps	Rob
Sulfur derivatives				
dimethyl disulfide	747			3.8
dimethyl trisulfide	974			0.3
Monoterpene hydrocarbons				
α-pinene	941	1.7		
sabinene	977	0.6		
β-pinene	982	2		
myrcene	993	2.9		
δ-3-carene	1013	1.2		
*p*-cymene	1028	1		
limonene	1032	34	0.6	1
γ-terpinene	1063	1.3		
Oxygenated monoterpenes				
fenchone	1089	1.2		
linalool	1101	2.3		
carvone	1244	1		
Non-terpene derivatives				
Non-terpene esters				
ethyl butyrate	804	1.4	0.5	
methyl valerate	826		1.6	
ethyl 2-methylbutyrate	842	0.8	5.6	
isopentyl acetate	877	0.6		
propyl butyrate	915	1.8	0.6	
isobutyl butyrate	956	0.7		
2-methylbutyl isobutyrate	1015	1.7	0.6	
methyl heptanoate	1027		0.5	
pentyl isobutyrate	1058		0.2	
methyl (*E*)-4-octenoate	1120		0.5	
methyl octanoate	1128	0.6	2.2	
hexyl isobutyrate	1152		0.2	
methyl nonanoate	1228		0.2	
Non-terpene aldehydes/ketones/acids				
isobutyric acid	772			79.5
butyric acid	799			1.4
furfural	834			5.1
2-methylbutanoic acid	846	2.6	4.7	6.3
2-heptanone	891	2.5	0.8	
6-methyl-5-hepten-2-one	987	1.2		
hexanoic acid	988		0.3	
octanal	1002	0.7		
2-nonanone	1093		0.2	
nonanal	1102	3.1	0.5	
8-methylnonanal	1172	0.6		
decanal	1205	1		
Non-terpene hydrocarbons				
styrene	897	7.1		
4-methyldecane	1059	0.7		
2-methyldecane	1064	0.6		
*n*-undecane	1100	0.8		
*n*-dodecane	1200	1.9		
Non-terpene alcohols/ethers/phenols				
1-hexanol	869	2.3		
4-heptanol	876			1.2
2-heptanol	897	5.6		
2-acetylfuran	916			0.2
4-octanol	979			0.2
Sulfur derivatives		0	0	4.1
Non-terpene hydrocarbons		11.1	0	0
Non-terpene aldehydes/ketones/acids		11.7	6.5	92.3
Non-terpene alcohols/ethers/phenols		7.9	0	1.6
Non-terpene esters subtotal		16.9	91.8	0.2
Total identified		96.8	98.9	99.2

a: Percentages obtained by FID peak area normalization. b: Linear retention indices (DB-5 column). Trace < 0.1% not reported.

**Table 2 molecules-25-03120-t002:** Phytochemical analysis of pulps, seeds decoction extracts and Rob.

Compounds	Pulps	Seeds	Rob
Anthocyanins ^x^	158.1 ± 0.7 ^b^	51.2 ± 0.1 ^a^	188.5 ± 5 ^c^
Phenolic ^y^	1.8 ± 0.1 ^a^	0.9 ± 0.1 ^a^	13 ± 0.8 ^b^
Flavonoid ^z^	0.6 ± 0.0 ^a^	0.6 ± 0.1 ^a^	2.4 ± 0.1 ^b^
Condensed tannins ^z^	0.3 ± 0.0 ^a^	0.2 ± 0.0 ^a^	1.2 ± 0.1 ^b^

x: as mg/100 g DW y: as Eg AC/100 g DW; z: as Eg C/100 g DW. Values not followed by the same letters (a, b and c), in the same line are significantly different for *p* < 0.05 (SPSS 22, ANOVA, post hoc test, Duncan).

**Table 3 molecules-25-03120-t003:** Summary of antioxidant activity evaluated by three methods (FRAP, DPPH and ABTS).

	DP	DS	Rob
DPPH	1.04 ± 0.0 ^c^	0.86 ± 0.0 ^a^	0.97 ± 0.1 ^b^
ABTS	3.49 ± 0.0 ^c^	1.61 ± 0.1 ^b^	0.94 ± 0.1 ^a^
FRAP	0.66 ± 0.0 ^b^	0.73 ± 0.1 ^c^	0.55 ± 0.0 ^a^

Values not followed by the same letters (a, b and c), in the same line, are significantly different for *p* < 0.05 (SPSS 22, ANOVA, post hoc test, Duncan).

**Table 4 molecules-25-03120-t004:** Analgesic activity of carob pulps (DP), seeds (DS) and Rob.

Groups	Times (min)	5	10	15	20	25	30	Total (Cumulative)
Control	Number of cramps	50 ± 3	151 ± 2	177 ± 5	125 ± 2	111 ± 2.3	62 ± 1.2	676 ± 3
Lysine	Number of cramps	5 ± 2	98 ± 6	60 ± 5	75 ± 3	79 ± 4.4	52 ± 2.9	369 ± 10 ***
Acetyl								
salicylate (200 mg/kg)	% inhibition	90 ± 1 ***	35.1 ± 3	66.1 ± 4 *	40.0 ± 2	28.8 ± 2.6	16.1 ± 1.3	45.40%
	Number of cramps	26 ± 3	97 ± 2	61 ± 3	49 ± 4	31 ± 3	23 ± 1	287 ± 4 ***
DP (50 mg/kg)	% inhibition	48 ± 1.7	35.8 ± 0.9	65.5 ± 1.7 *	60.8 ± 2.5	72 ± 1.3 **	62.9 ± 0.4 *	57.50%
DP (100 mg/kg)	Number of cramps	23 ± 2	89 ± 4	47 ± 2	41 ± 2	29 ± 3	24 ± 2	253 ± 5 ***
	% inhibition	54.0 ± 1.2	41.1 ± 2,4	73.5 ± 1 **	67.2 ± 0.7 *	73.9 ± 1.4 **	61.3 ± 0.7 **	62.60%
DP (150 mg/kg)	Number of cramps	34 ± 4	81 ± 5	63 ± 5	46 ± 4	23 ± 2	12 ± 3	249 ± 10 ***
	% inhibition	52.00 ± 1.1	46.36 ± 3	64.4 ± 2.6 *	63.2 ± 2.2 *	79.28 ± 0.9 **	80.64 ± 1.3 **	63.20%
DP (200 mg/kg)	Number of cramps	35 ± 3	96 ± 5	43 ± 3	39 ± 3	22 ± 1	10 ± 2	226 ± 5 ***
	% inhibition	30 ± 1.7	49 ± 1.4	75.7 ± 1.7 **	68.8 ± 1.9 *	80.2 ± 0.6 **	83.9 ± 0.6 **	66.60%
	Number of cramps	10 ± 1	73 ± 1	48 ± 1	54 ± 2	27 ± 2	30 ± 1	242 ± 4 ***
DS								
(50 mg/kg)	% inhibition	87.3 ± 2 ***	51.7 ± 2	72.9 ± 2.5 **	56.8 ± 4.5	75.7 ± 2.6 **	51.6 ± 0.9	64.20%
DS (100 mg/kg)	Number of cramps	13 ± 1	56 ± 3	30 ± 2	19 ± 1	17 ± 1	16 ± 2	151 ± 6 ***
	% inhibition	83.54 ± 2.5 **	63 ± 6.3	83 ± 3 **	84.8 ± 2.1 **	84.7 ± 2.3 **	74.2 ± 2.8 **	77.70%
DS (150 mg/kg)	Number of cramps	35 ± 2	64 ± 1	51 ± 1	23 ± 1	18 ± 1	15 ± 1	206 ± 4 ***
	% inhibition	55.7 ± 3.5	57.6 ± 2.6	71.2 ± 2.7 **	81.6 ± 3.1 **	83.8 ± 3.7 **	75.8 ± 1.6 **	69.50%
DS (200 mg/kg)	Number of cramps	41 ± 1.4	72 ± 1	48 ± 2	35 ± 1	23 ± 1	6 ± 1	225 ± 4 ***
	% inhibition	48.1 ± 3.4	52.3 ± 2.9	72.9 ± 3.2 **	72 ± 2 **	79.3 ± 2 **	90.3 ± 1.1 ***	66.70%
	Number of cramps	24 ± 1.1	46 ± 1	38 ± 1	41 ± 1	27 ± 0.6	20	196 ± 8 ***
Rob (50 mg/kg)	% inhibition	69.6 ± 7 *	69.5 ± 4.1	78.5 ± 3 **	67.2 ± 3.8 *	75.7 ± 3 **	67.7 ± 2.6 *	72.20%
	Number of cramps	25 ± 2.8	53 ± 2	29 ± 2	25 ± 2	16 ± 1.2	7 ± 1	155 ± 6 ***
Rob (100 mg/kg)	% inhibition	68.4 ± 1.6 *	62.9 ± 4.35 *	83 ± 2.8 **	84.8 ± 3.4 **	84.7 ± 1.9 **	74.2 ± 0.8 **	78%
Rob (150 mg/kg)	Number of cramps	28 ± 1	62 ± 2	47 ± 2	43 ± 2	31 ± 1	10 ± 1	221 ± 3 ***
	% inhibition	64.6 ± 0.8 *	58.9 ± 4	73.5 ± 2.6 **	65.6 ± 1.2 *	72.1 ± 1.3 **	83.9 ± 1.97 **	68.70%
Rob (200 mg/kg)	Number of cramps	32 ± 1	73 ± 2	46 ± 1	43 ± 0.5	33 ± 0.5	10 ± 0.8	237 ± 4 ***
	% inhibition	63.5 ± 2.7 *	51.5 ± 2.6	75.6 ± 2.9	67.4 ± 1.9 *	71.9 ± 3.6 **	82.6 ± 1.5 **	66.40%

Results are expressed as mean ± SD for 6 rats in each group. The asterisks ** and *** indicate significant differences respectively at *p* < 0.01 and *p* < 0.001 relative to control (ANOVA, post hoc LSD test) DP: decoction pulps, DS: decoction seeds.

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
