# Peer review of "Biological Activities of Aqueous Extracts from Carob Plant (Ceratonia siliqua L.) by Antioxidant, Analgesic and Proapoptotic Properties Evaluation"

_molecules, 2020, doi:10.3390/molecules25143120_

Round 1
Reviewer 1 Report
In this paper Author describe the volatile compounds profile and phytochemical content of Ceratonia siliqua L. Fifty different components have been identified. Among them, three constituents are shared i.e. 2-methlybutanoic acid, methyl hexanoate and limonene by different common carob preparations: pulp decoction (PD), seeds decoction (SD) and Rob, a sweet syrup extracted from the pulp of the carob pod. Antioxidant activity of the extracts was evaluated using three methods (DPPH, ABTS and FRAP) producing a dose-dependent response. The IC50, in FRAP assay, gave the lowest values (0.66±0.01, 0.73±0.05 and 0.55±0.00 mg/ml PD, SD and Rob, respectively). The nociception assay, after intraperitoneal injection of acetic acid in mice, demonstrated that Rob, pulp and seeds decoction extracts showed an efficient inhibition. According to these data, the seeds carob extract exhibits potent radical scavenging properties, resulting in analgesic activity. Furthermore, the proapoptotic activity of SD against three human cell line (THP-1, MCF-7 and LOVO) were evaluated, showing a significantly stronger proapoptotic activity on colon cancer (LOVO) than on the other cell lines.
All these results evidenced that carob pulps, seeds and Rob may be implicated in several pathways, contributing to antioxidant activity by mean of their bioactive components. Most important, analgesic action of carob extracts seems to be an attractive new approach for pain treatment.
The paper is well written and of good scientific interest. The pharmacological analysis is thorough, although in my opinion a little confusing. In fact, the different pharmacological activities tested are not very correlated.
In detail, Authors, if possible, should add:
- Test on antioxidant action in vivo cells, to verify the inhibitory action on ROS in vivo.
- Anti-inflammatory test in vivo: in fact, in my opinion, the anti-inflammatory activity, compared to the analgesic one, is more related to the anti-tumor action.
Minor revisions:
- Delete figure 1 and figure 2 because not fundamental.
- Figure 3 and Figure 4: please insert unit of measure in the graphics.
- Page 15, lines 364-366: these data are not clear. Are Western blot analysis inserted in ref 53? Please specify.
- References, ref 1: please check
Author Response
Q: Question; R: Response
Reviewer 1:
Q1- Test on antioxidant action in vivo cells, to verify the inhibitory action on ROS in vivo.
Q2- Anti-inflammatory test in vivo: in fact, in my opinion, the anti-inflammatory activity, compared to the analgesic one, is more related to the anti-tumor action.
R1-R2: anti-inflammatory activity has been performed since and the corresponding results are part of an article in redaction.
Q3- Delete figure 1 and figure 2 because not fundamental.
R3- Done, figures have been ignored, and all figures renumbered as well.
Q4- Figure 3 and Figure 4: please insert unit of measure in the graphics.
R4- Done, X axis: Concentration (mg/ml); Y axis: Percentage of scavenging activity (%)
Q5- Page 15, lines 364-366: these data are not clear. Are Western blot analysis inserted in ref 53? Please specify.
R5- The carob extract pods has shown a high inhibition on human colon cell lines growth, but the effect on DNA-synthesis which was determinate with the single cell microgel electrophoresis assay was less pronounced in HT29 adenocarcinoma than LT97 adenoma cells, certainly due to different growth kinetics [52].
Q6- References, ref1: please check
R2- Done: Liu, R. H., Dietary bioactive compounds and their health implications. J Food Sci 2013, 78 Suppl 1, A18-25.doi: 10.1111/1750-3841.12101
Reviewer 2 Report
- Please provide the full spelling of PD and SD in the main text.
- Please unify the writing of pulp decoction (PD), seeds decoction (SD) and Rob in the main text.
- Please provide the animal species and the IACUC approval number.
- Please the pro-apoptotic data of positive control celastrol.
- Please fix the annotation of table 2 in line 249 -251, page 9.
- Please correct the description in line 291 – 292, page 11. The description is different with Table 2.
- Three samples (PD, SD and Rob) have different activities against ABTS, DPPH and ferrous radical. Why?
- Please explain the reason for choosing statistical methods in antioxidant and pro-apoptotic experiments.
- Please change table 4 with one figure. Only cumulative count must be showed in the figure.
- From the results of figure 5 – 7, the proapoptotic effects of three samples are not good.
- Please show the data of PI and Annexin-V-FITC double stain.
- The description in line 375 -378, page 15 must be corrected. According to figure 7, the effects of SD at 0.5 – 1.0 mg/mL have reached a plateau.
Author Response
Reviewer 2:
Q1- Please provide the full spelling of PD and SD in the main text.
R1- lines 238-239 The level of flavonoids in pulp decoction (PD) and seeds decoction (SD) extracts is around 5 times lower, with 0.6±0.0 and 0.6±0.1 g EC/100 g, respectively.
Q2- Please unify the writing of pulp decoction (PD), seeds decoction (SD) and Rob in the main text.
R2- Done
Q3- Please provide the animal species and the IACUC approval number
R3- was detailed accordingly in page 3 line 129.
Swiss albino mice of both sexes weighing18–25 g, were obtained from Pasteur Institute (Tunis, Tunisia). The animals were housed under standard laboratory conditions with a 12-h light–dark cycle at constant temperature (22 ± 2 °C) and humidity (55 ± 5%) and kept on commercial diet and tap water provided ad libitum. The animals were handled according to the guidelines of the Tunisian Society for the Care and Use of Laboratory Animals, and the study was approved by the University of Monastir Ethical Committee (approval # CER-SVS 007/2020 ISBM).
Q4- Please the pro-apoptotic data of positive control Celastrol.
R4- here are mean±sd of % of pro-apoptotic population for leukemia cells (THP1), breast cancer cells (MCF-7) and adenoma colon cells (LOVO):
|
Cells |
THP-1 |
MCF-7 |
LOVO |
|
Untreated cells |
1.2 ± 0.1 |
2.3 ± 0.4 |
0.9 ± 0.2 |
|
Positive control (Celastrol) |
85.9 ± 0.9 |
96.7 ± 0.2 |
99.99 ± 0.03 |
Q5 : Please fix the annotation of table 2 in line 249 -251, page 9.
R5 : Table 2.Phytochemical analysis of pulps, seeds decoction extracts and rob.
|
Compounds |
Pulps |
Seeds |
Rob |
|
Anthocyaninsx |
158.1±0.7b |
51.2±0.1a |
188.5±5c |
|
Phenolicy |
1.8±0.1a |
0.9±0.1a |
13±0.8b |
|
FlavonoidZ |
0.6±0.0a |
0.6±0.1a |
2.4±0.1b |
|
Condensed tanninsZ |
0.3±0.0a |
0.2 ±0.0a |
1.2 ±0.1 b |
x: as mg/100g DW y: as EAG /100g DW; z: as EC /100g DW. Values not followed by the same letters (a, b and c), in the same line are significantly different for p<0.05 (SPSS 22, ANOVA, post hoc test, Duncan).
Q 6- Please correct the description in line 291 – 292, page 11. The description is different with Table 2.
R6- Page 11, lines 293-294
Thus, Rob reveals potent antioxidant activity as shown by FRAP and ABTS and a high antiradical capacity. This antioxidant activity may be attributed to the levels of total phenols, flavonoids, condensed tannin and anthocyanin (13.0±0.8 g EAG, 2.4±0.1 g EC, 1.2±0.1 g EC and 188.5±5.0 DW, respectively).
Q7- Three samples (PD, SD and Rob) have different activities against ABTS, DPPH and ferrous radical. Why?
R7 – According to the complexity of the oxidation processes and the diverse nature of antioxidants, knowing that these are both hydrophilic and hydrophobic components. This implies that there is no universal and reliable method for the evaluation of antioxidant activity. This is why it is necessary to combine the responses of different and complementary tests aimed at having a precise idea on the antioxidant capacity of the sample to be studied.
Q8- Please explain the reason for choosing statistical methods in antioxidant and pro-apoptotic experiments.
R8- According to the normality test, the distribution of this study followed Gaussian law. It is for that reason that we used parametric tests (ANOVA).
Q9- Please change table 4 with one figure. Only cumulative count must be showed in the figure.
R9- As in the main text we mentioned the reduction of writhing reflexes and not the cumulative count the table remains important in our opinion as a graph tab would be less informative.
Figure x : Cumulative count of writhing reflexes of carob pulps (PD), seeds (SD) and Rob. Results are expressed as mean± SD for 6 rats in each group. The asterisks**,*** indicate significant difference respectively at P<0.01 andP< 0.001 relative to control (ANOVA, post hoc LSD test) PD: pulp decoction, SD: seeds Decoction, ASL : Lysine acetly silicilate
Q10- From the results of figure 5 – 7, the proapoptotic effects of three samples are not good.
R10- To obtain a full response in apoptosis induction, the concentration of the extracts to reach maximum effect should have been higher. The conclusion is that extracts induce differently depending the tested cell type, a phenomenon described as phenotypic specificity, and that is exactly the point we wanted to highlight here, so that "not god" seems unappropriated for our point of view.
Q11- Please show the data of PI and Annexin-V-FITC double stain.
R1: 1the sentence explaining the origin of % of apoptotic cells, have been added in the text i.e.: % of apoptotic cells = Annexin V positive population + Annexin V/PI positive population.
Q12- The description in line 375 -378, page 15 must be corrected. According to figure 7, the effects of SD at 0.5 – 1.0 mg/mL have reached a plateau.
R12- we did not comment our work here, as line 375-378 concern the description of another work (ref 12 and 54).
- Klenow, S.; Jahns, F.; Pool-Zobel, B. L.; Glei, M., Does an extract of carob (Ceratonia siliqua L.) have chemopreventive potential related to oxidative stress and drug metabolism in human colon cells? J Agric Food Chem 2009, 57, (7), 2999-3004
- Roseiro, L. B.; Duarte, L. C.; Oliveira, D. L.; Roque, R.; Bernardo-Gil, M. G.; Martins, A. I.; Sepúlveda, C.; Almeida, J.; Meireles, M.; Gírio, F. M., Supercritical, ultrasound and conventional extracts from carob (Ceratonia siliqua L.) biomass: Effect on the phenolic profile and antiproliferative activity. Industrial crops and products 2013, 47, 132-138
Reviewer 3 Report
Dear Authors
There is no information about experimental animals and their origin - line 127 and 137.
There is no information about the approval of the Bioethics Committee for these studies.
All literature must have DOI.
References 6, 7, 17, 18, 20,21, 22, 25, 28, 29, 32 and 37 are too old and must be changed.
Author Response
Reviewer 3:
Q1- There is no information about experimental animals and their origin - line 127 and 137.
R1- Added in the text line 129-131
Q2- There is no information about the approval of the Bioethics Committee for these studies
R2- Added in the text line 133-134
Q3- All literature must have DOI
R3- We used Endnote to format our References using the given "molecule" template so that the DOI never appears according to this format.
Q4- References 6, 7, 17, 19, 25, 28, 29, 32 and 37 are too old and must be changed.
R4: Concerning references 6, 7 noted as "too old" (1989, 1995), we do not agree, since these authors made the identification and the systematic classification of carob tree in Tunisia and so for the majority of Tunisian studies refer to these authors. By the way, having myself (CDM) started to publish in 1983, I found the "too old" notion to be a bit offending. ;-)
More seriously, the following references have all been updated by more recent ones:
Line 114:
- V. I. Babushok, P. J. Linstrom, I. G. Zenkevich, Retention indices for frequently reported compounds of plant essential oils, Journal of Physical and Chemical Reference 2011, Data, Vol. 40, art. n°043101, https://doi.org/10.1063/1.3653552
- P.J. Linstrom and W.G. Mallard, Eds., NIST Chemistry WebBook, NIST Standard Reference Database Number 69, National Institute of Standards and Technology, Gaithersburg MD, 20899, https://doi.org/10.18434/T4D303,
Line 119: The total flavonoids content was determined by a colorimetric method according to (Reis et al., 2011)[25]
- Reis, F.S., Heleno, S.A., Barros, L., Sousa, M.J., Martins, A., Santos-Buelga, C., Ferreira, I.C.F.R., 2011. Toward the Antioxidant and Chemical Characterization of Mycorrhizal Mushrooms from Northeast Portugal. J. Food Sci. 76, C824–C830. https://doi.org/10.1111/j.1750-3841.2011.02251.x
Line 126: All measurements were carried out according to Pinela and al. (2012) [27] and to Barros et al., (2011)[28], respectively
- Barros, A.I.R.N.A., Nunes, F.M., Gonçalves, B., Bennett, R.N., Silva, A.P. Effect of cooking on total vitamin C contents and antioxidant activity of sweet chestnuts (Castanea sativa Mill.). Food Chem. 2011, 128, 165–172. https://doi.org/10.1016/j.foodchem.2011.03.013
Line131: The mortality rate within a 48 hours period was determined as LD50 estimated according to the method of Wang et al.( 2007) [29]
- Wang, J., Zhou, G., Chen, C., Yu, H., Wang, T., Ma, Y., Jia, G., Gao, Y., Li, B., Sun, J., 2007. Acute toxicity and biodistribution of different sized titanium dioxide particles in mice after oral administration. Toxicol. Lett. 168, 176–185. https://doi.org/10.1016/j.toxlet.2006.12.001
Line 217: In this context,Krokou et al. (2019) [32] found more than 50 volatile components were identified in carob powder, mainly acids, esters and aldehydes/ ketones. The most abundant compound in the volatile profile is isobutyric acid which gives the sweet and buttery flavor of carob.
- Krokou, A., Stylianou, M., Agapiou, A. Assessing the volatile profile of carob tree (Ceratonia siliqua L.). Environ. Sci. Pollut. Res. 2019 26, 35365–35374. https://doi.org/10.1007/s11356-019-04664-7
Line 245: According to other works, carob pods show variations, according to the used solvent system, ranging from 7.1 to 382.0 mg EGA of phenolic content and 98.7 mg EC of flavonoids [37]
- Goulas, V., Georgiou, E. Utilization of Carob Fruit as Sources of Phenolic Compounds with Antioxidant Potential: Extraction Optimization and Application in Food Models. Foods 2019, 9, 20. https://doi.org/10.3390/foods9010020
Round 2
Reviewer 1 Report
the major part of my suggestions has been done by the authors; in my opinion paper can be accepted.
Reviewer 2 Report
The manuscript has been significantly improved and can be published now.
Reviewer 3 Report
Thank you for explaining. I have no more comments